depression; HIV; sub-Saharan Africa; group interpersonal therapy

**Corresponding author:**
Charlotte Bernard;
Email: charlotte.bernard@u-bordeaux.fr

# Management of depression in people living with HIV/AIDS in Senegal: Acceptability, feasibility and benefits of group interpersonal therapy

Charlotte Bernard[1] 🆔, Hélène Font[1], Salaheddine Ziadeh[2,3], Judicaël M. Tine[4], Abibatou Diaw[5], Ibrahima Ndiaye[6], Oumar Samba[6], Thierry Bottai[7], Laurent Jacquesy[8], Helena Verdeli[2], Ndeye F. Ngom[5], François Dabis[1], Moussa Seydi[4], Nathalie de Rekeneire[9] and The IeDEA West Africa Cohort Collaboration

[1]University of Bordeaux, National Institute for Health and Medical Research (INSERM) UMR 1219, Research Institute for Sustainable Development (IRD) EMR 271, Bordeaux Population Health Centre, Bordeaux, France; [2]Global Mental Health Lab, Teachers College, Columbia University, New York, NY, USA; [3]Faculté de Santé Publique, Université Libanaise, Sidon, Lebanon; [4]Service des Maladies Infectieuses et Tropicales, CHNU de Fann, Dakar, Senegal; [5]Centre de Traitement Ambulatoire, CHNU de Fann, Dakar, Senegal; [6]Service de Psychiatrie, CHNU de Fann, Dakar, Senegal; [7]Pôle de Psychiatrie, CH de Martigues, Martigues, France; [8]Psychiatre Indépendant, Président de CREATIP, Annecy, France and [9]Institut Pasteur du Cambodge, Phnom Penh, Cambodia

## Abstract

Depression is highly prevalent in people living with HIV (PLWH) and has negative consequences for daily life and care. We evaluated for the first time the acceptability, feasibility and benefits of group interpersonal therapy (IPT), combined with a task-shifting approach, to treat depression in PLWH in Senegal. PLWH with depression received group IPT following the World Health Organization protocol. Acceptability and feasibility criteria were defined from the literature data. The PHQ-9, the WHODAS, and the 12-item-stigma scale were used, pre- and post-treatment, including a 3-month follow-up, to assess depressive symptom severity, functioning and stigma, respectively. General linear mixed models were used to describe changes in outcomes over time. Of 69 participants, 60 completed group IPT. Refusal to enroll and dropout rates were 6.6 and 12.7%, respectively. Ninety-seven percent of participants attended at least seven out of eight sessions. Patients and facilitators endorsed group IPT, with willingness to recommend it. Depressive symptoms and disability improved drastically and sustainably. We showed that group IPT is well accepted and feasible in Senegal as treatment for depression in PLWH. Combined with a task-shifting approach, it can narrow the gap in mental health treatment. Implementation may be enhanced by refining patient identification procedures and increasing treatment accessibility.

## Impact statement

Depression is highly prevalent in people living with HIV (PLWH) and has negative consequences for patient daily life and healthcare. Yet, depression remains mostly underdiagnosed and undertreated in sub-Saharan Africa. In resource-limited settings, such as Senegal, the World Health Organization recommends task-shifting and psychotherapeutic interventions to treat depression. The present study is a first evaluation of the acceptability and feasibility of group interpersonal therapy (IPT) to treat depression in PLWH in Senegal. Groups were facilitated by trained social workers and community health workers. Our results showed that group IPT was well accepted and feasible in this country as treatment for depression in PLWH and that it drastically decreased depression severity and associated disability. As group IPT was administered by non mental health specialists, its success can narrow the gap in mental health treatment. In implementation, some treatment barriers to group IPT's success and sustainability were identified (e.g., refining identification procedures and bringing services closer to patients). We concluded that group IPT was a promising intervention that could close the mental health treatment gap in Senegal, and we discussed how some measures might transform group IPT into a standard of care for depressed PLWH in the country and, possibly, beyond.

## Background

Depression is highly prevalent in sub-Saharan Africa (SSA) and is the most prevalent psychiatric disorder in people living with HIV (PLWH) (Bernard et al., 2017). It is associated with negative

consequences for the quality of life (Abas et al., 2014) and HIV outcomes (i.e., decreased adherence to antiretroviral therapy (ART), slower increase in CD4 count and rapid progression to AIDS stage) (Berhe and Bayray, 2013; Memiah et al., 2014; Wroe et al., 2015). Despite its negative impact on the continuum of care, depression remains often underdiagnosed and undertreated in SSA (Berhe and Bayray, 2013; Abas et al., 2014).

Evaluating promising models of integrated mental health and HIV care was reported as a research priority in the "treat all" area (Parcesepe et al., 2018a, 2018b). Indeed, significant gaps exist in the management of mental disorders in HIV care settings in low- and middle-income countries (LMICs) (i.e., limited screening availability, non systematic screening, limited treatment availability), limiting access to care and treatment (Parcesepe et al., 2018a, 2018b). The World Health Organization (WHO) recommends the use of psychological interventions as first-line treatment in LMIC. However, there are many challenges to implementing evidence-based psychotherapeutic treatments, such as lack of mental health specialists, limited resources dedicated to mental health care and poor integration of mental health services (Lancet Global Mental Health Group et al., 2007; WHO, 2010; Patel et al., 2011). In this context, the WHO also encourages task-shifting, an approach that involves the training of non mental health specialists to provide mental health care under the guidance of specialists (Lancet Global Mental Health Group et al., 2007; WHO, 2008). Task-shifting is widely acknowledged as an optimal method to sustain the implementation of psychological interventions (Patel and Thornicroft, 2009; Patel et al., 2010).

The WHO also specifically recommends group interpersonal therapy (group IPT) as a first-line intervention to treat depression in LMICs (WHO, 2016). Developed in the 1970s by Klerman and Weissman (Klerman et al., 1974) as individual treatment for depression, Interpersonal psychotherapy (IPT) has been successfully adapted to different psychiatric conditions, intervention modalities, age groups, and settings around the world. IPT is based on the idea that depressive symptoms are triggered by interpersonal problems in one or more of the following areas: disagreement or conflict; life changes (negative or positive); grief; loneliness or social isolation (Markowitz and Weissman, 2004; Weissman, 2006; Weissman et al., 2018). Patients come to see their depression as a treatable condition within an interpersonal context that can be changed or managed effectively with interpersonal competencies and strategies they learn and apply during the course of treatment; ultimately, they come to resolve the problem area and avert a future relapse. IPT's effectiveness in treating depression has been reported in two systematic reviews (Cuijpers et al., 2016, 2011) and with various patient populations, including PLWH (Markowitz et al., 1998). IPT was first applied as a group intervention to treat adult depression in Uganda, with encouraging outcomes (Bolton et al., 2003; Verdeli et al., 2003a; Bass et al., 2006). Authors reported the benefits of using a group modality (e.g., wider patient coverage and lower implementation costs) (Verdeli et al., 2003a). In a US study with PLWH population, IPT appeared to be the most effective method to treat depression compared to cognitive behavioral and supportive psychotherapy, and equally effective to antidepressants combined with supportive psychotherapy (Markowitz et al., 1998). In South Africa, receiving an HIV diagnosis, bereavement (particularly multiple losses due to HIV/AIDS), conflict with family members, and financial stress emerged as triggers for depression (Bana et al., 2009 cited in Petersen et al., 2012). These triggers are consistent with the problem areas associated with depression in IPT, arguably enhancing its suitability for PLWH. The study also

reported a significant improvement in depressive symptoms in PLWH treated with a modified form of group IPT, compared to a control group (Petersen et al., 2014).

Implementation of psychological treatments, however, could be different from one population (or country) to another and might need some cultural adaptation. It also requires specialized training and can increase the workload on staff (Patel et al., 2011; Padmanathan and De Silva, 2013; Bach-Mortensen et al., 2018). Therefore, examining the acceptability and feasibility of such treatments is crucial, especially in the context of its dissemination through the healthcare system of an entire country. Such evaluation is particularly pertinent in the context of HIV/AIDS, where the "HIV and depression" double burden and high related stigma could reduce adherence to care. A recent study in Ethiopia concluded that group IPT was feasible and acceptable in PLWH (Asrat et al., 2020, 2021). Nonetheless, to our knowledge, there are no data available in West Africa or, more specifically, Senegal.

The Senegalese health system has an extensive experience in psychiatry, but suffers from the scarcity of mental health specialists and inequalities in access to mental health care (Petit, 2022). In this context, we aimed to assess the acceptability and feasibility of group IPT combined with a task-shifting approach (i.e., group IPT was delivered by trained social or community health workers) to treat PLWH with depression in Senegal. A second aim was to evaluate whether after group IPT, disability and HIV-related stigma decreased compared to baseline scores.

## Methodology

### Study design

The present study is a part of an ancillary study project within the West Africa network of the International epidemiological Databases to Evaluate AIDS (IeDEA) of the US National Institutes of Health (https://www.iedea.org/regions/west-africa/) (Egger et al., 2012). The study took place between October 2019 to March 2022 (end of 3-month follow-ups), with an interruption of 4 months due to the COVID-19 pandemic. In this cross-sectional descriptive study, we compared pre- and post-intervention data (depressive symptoms, functioning and HIV-related stigma) in PLWH treated for depression with group IPT. Assessments were conducted pre-intervention, post-intervention, and at 3-month follow-up.

### Setting

The present study was conducted at the Fann National University Hospital Center (FNUHC) in Dakar, the capital and largest city in Senegal (population ~ 4,000,000). The FNUHC is at the top of the healthcare pyramid and is a pioneer in the field of psychiatry in West Africa. The study involved two service units that provide treatment and follow-up for PLWH (~ 2,500 patients): (1) Infectious and Tropical Diseases unit; (2) Outpatient Treatment Center. The psychiatry department contributed two Senegalese psychiatrists to confirm the diagnosis of depression in study participants, and to assist with the management of severe study cases if necessary.

### Study population

The source population of our study was PLWH followed in HIV care centers. PLWH were voluntarily screened for depression by a designated clinician at the time of their regular HIV clinical visit to the hospital. To be included, PLWH had to be adults aged ≥20 years,

on ART, and identified as "depressed" after being screened with the Patient Health Questionnaire (PHQ-9) (Kroenke et al., 2001) by one clinician who works in one of the participating sites on the project. In a second step, patients with a PHQ-9 score ≥ 5 saw a study psychiatrist to confirm during a clinical interview whether they had met the diagnostic criteria for depression. The exclusion criteria were: hospitalization and/or medical emergency; diagnosis with a psychiatric illness other than depression; vision or hearing impairment that would seriously hinder group interactions; and imminent suicide risk. Patients with active suicidal risk were referred to a psychiatrist for further evaluation and intervention.

### Ethical approval

The research was conducted in accordance with the Helsinki Declaration. Ethical approval was obtained from Senegal's ethics committee: Conseil National d'Ethique de la Recherche en Santé (CNERS). All participants gave their written consent prior to participation. Informed consent was obtained in a private room typically reserved for medical visits. All collected data were anonymized.

### Group IPT

The intervention was implemented following the WHO manual guidelines, which had been translated to French by our team (https://www.who.int/publications/i/item/WHO-MSD-MER-16.4). Group IPT was delivered over the course of eight weekly group sessions, preceded by one pre-group individual meeting. The eight sessions covered the three phases of treatment: initial, middle, and termination. Groups were gender-specific (facilitator included) and consisted of six patients and one group-facilitator each. Therapy was provided in French and/or in Wolof, Senegal's national language. The expression "naxaru xol" (literally, "loss of appetite for doing things" in Wolof) was chosen to refer to depression. The term "depression" is less acceptable culturally, as it has connotations with madness.

Group facilitators (i.e., trained staff who lead group IPT groups) were Senegalese social workers ($N$ = 3) or community health workers ($N$ = 1). They had no specialized training on depression or its treatment with psychotherapy. Group facilitators were trained in IPT in two cycles. First, at the end of October 2018, a 3-day training in individual IPT was conducted by two French psychiatrists (T.B., L.J.) from the Cercle de Recherche et d'Etudes Appliquées à la Thérapie InterPersonnelle (CREATIP) association. Following this training, trainees had to conduct IPT with two patients to establish competence. Second, in March 2019, a 5-day training in group IPT was conducted by a master trainer (S.Z.), who is a clinical psychologist. To establish clinical proficiency, the training was followed by two rounds of supervised practice, led by the master trainer, from March to October 2019, where trainees first worked in pairs (cofacilitators) then alone (single facilitator) to provide group IPT to PLWH. Following this training, the study's implementation phase began in October 2019 in the sites described above. In the course of the study, one of the trained facilitators – a social worker – moved to another service and stopped group facilitation. Group facilitators were supervised by the master trainer (S.Z.) throughout the study phase.

### Implementation outcomes

#### Acceptability

Acceptability is defined as how the patient perceives or reacts to treatment (Craig et al., 2008; Patel et al., 2011; Proctor et al., 2011).

In the present study, acceptability was assessed by the refusal rate (number of patients with depression who refused to participate in group IPT/number of patients with depression who were invited to participate), and the drop-out rate (number of patients who were included but did not continue to attend group IPT sessions/number of patients who agreed to participate) (Purcell et al., 2007). Patient satisfaction with this therapy was evaluated using four questions adapted from the Consumer Satisfaction Questionnaire (CSQ-8) (Kapp et al., 2014). Specifically, patients were asked to rate (on a 4-point Likert scale) the quality of the service they received, to see whether it was in line with their expectation and the extent to which the treatment met their needs. They also indicated their general level of satisfaction.

#### Feasibility

Feasibility is defined as the extent to which an intervention can be provided successfully for a specific population (Bowen et al., 2009; Proctor et al., 2011). The feasibility was assessed according to different indicators: attendance of therapy sessions, risk of suicide during group IPT delivery, duration of treatment in practice (i.e., 8 weeks or more) and the time between confirmation of diagnosis and starting group IPT. We also asked facilitators and patients whether they would recommend this therapy to others patients. Finally, facilitators were asked to evaluate the feasibility of group IPT, with a self-administered questionnaire that tapped their satisfaction with group IPT, the ease of its implementation in their daily practice and challenges they might have encountered.

### Depressive symptoms, functioning and HIV-related stigma

Depressive symptoms, functioning and HIV-related stigma were assessed by clinicians during follow-up visits (i.e., at the end of group IPT and 3 months later).

The PHQ-9 was used to evaluate the severity of depressive symptoms. Each item was rated on a 4-point Likert scale ranging from 0 to 3. A total score on the PHQ-9 was obtained by summing up the scores on the individual items and could range from 0 to 27. The total score indicated the severity of depressive symptoms: mild (5–9), moderate (10–14), moderately severe (15–19), or severe (≥20). The PHQ-9 is widely used in studies conducted with PLWH in SSA (Wagner et al., 2011; Belenky et al., 2014; Endeshaw et al., 2014; Musisi et al., 2014; Asangbeh et al., 2016) and is recommended for use by the mental health work group of the IeDEA Cohort collaboration (NIMH).

The World Health Organization Disability Assessment Schedule (WHODAS-12) was used to measure general health and disability (Ustün et al., 2010). This 12-item measure uses a 5-point Likert scale (i.e., no difficulty, mild difficulty, moderate difficulty, severe difficulty, extreme difficulty/"cannot do") to assess functioning across six domains (i.e., cognition, mobility, self-care, getting along, life activities, and participation in community activities). A total score is obtained by summing up item scores and could range from 12 to 60, with higher scores denoting more loss of function. The WHODAS has shown good psychometric properties and was used in several countries (Federici et al., 2017), particularly with people living with mental illness or chronic disease in Ethiopia (Habtamu et al., 2016), Nigeria (Igwesi-Chidobe et al., 2020), and Ghana (Badu et al., 2021).

The 12-item short version of the HIV Stigma Scale was also used to assess HIV-related stigma (Reinius et al., 2017). The measure includes 4 stigma subscales (i.e., personalized stigma, disclosure concerns, concerns with public attitudes and negative self-image) of three items and uses a 4-point Likert scale ranging from "strongly

agree" to "strongly disagree" to generate a total score. Subscale scores have a possible range of 3–12, with higher scores reflecting more stigma. This short version was created from the Berger Scale (40-item version). It was often used with PLWH in sub-Saharan Africa, including Nigeria (Onyebuchi-Iwudibia and Brown, 2014; Secor et al., 2015), Kenya (Kaai et al., 2010) and Cameroon (Ajong et al., 2018). Shorter versions were validated in Ethiopia (Feyissa et al., 2012) and Tanzania (Ramos et al., 2018).

### Statistical analysis

The characteristics of the sample were described using median and interquartile range (IQR) for continuous variables and counts and proportion for categorical variables. A comparison of baseline characteristics according to follow-up status (i.e., group IPT completed or not) was performed using the Wilcoxon rank test for continuous variables, and Chi-2 or Exact Fisher Test for categorical variables depending on the tests' conditions of applications.

Significant improvement of depressive symptoms, disability and HIV-related stigma scores were assessed between (1) inclusion and the end of group IPT and (2) between the end of group IPT and 3 months' post-treatment, using a general linear mixed model (Tukey–Kramer adjustment) ($p < 0.05$). Description of the different scores is presented with mean and standard deviation for each visit. The evolution of the PHQ-9 symptoms is presented in an alluvial plot (Rosanbalm, 2015), according to symptom severity. Patient Satisfaction Questionnaire responses and the opinions of the facilitators on feasibility are shown using stacked bars.

## Results

### Flow chart

In total, 559 PLWH were screened for depression using PHQ-9 and 165 had a PHQ-9 score ≥ 5 (Supplementary Figure S1). Of these 165 patients, 79 had a diagnosis of depression subsequently confirmed by the study's psychiatrist. Three patients had to be excluded from the study: one had received individual IPT and the other two lived too far away from the clinic and required several hours to commute. Participation in group IPT was offered to 76 patients.

Of the patients invited to participate in the study, two refused to participate in group IPT and three accepted but did not participate in any session, thus limiting the number of participants to 71. Subsequently, the data of two patients were excluded from analysis due to incorrect completion in their consent to participate. Therefore, a total of 69 patients were included in the present study.

In total, 12 IPT groups were formed to treat the study participants. In the course of treatment, nine patients dropped out. One patient completed the initial pre-group meeting, but did not participate in any group sessions (she was subsequently invited to join another group, which she accepted but did not attend). In contrast, 60 patients completed their group IPT treatment. No significant baseline differences were observed between patients who dropped out and those who completed their treatment (Supplementary Table S1). The 69 PLWH attributed their depression to one or more interpersonal problem areas, with primary problems (i.e., those that ranked first) distributed as follows: life changes (52.2%), conflict (36.2%), grief (10.2%) and social isolation (1.4%).

Among the patients who completed group IPT ($N = 60$), 93.3% attended their follow-up visit at the end of group IPT and 73.3% came to their 3-month post-treatment follow-up. Two patients did not attend their initial follow-up visit at the end of group IPT, but came to their 3-month post-treatment follow-up.

### Characteristics of the sample

Sixty-nine PLWH with depression were included in the study. The median age was 45 (interquartile range (IQR): 39–53) (Table 1); 50.7% were female; 43.5% lived alone and 42.0% were unemployed. About a quarter reported that they had not shared their HIV status with anyone (26.1%). A majority reported financial difficulties in the 6 months prior to starting group IPT (82.6%). No patient reported hazardous drinking or drug use. The median duration of HIV infection was 8 years (IQR 2–13), and 95.6% of participants reported 100% ART adherence. None of the participants were prescribed antidepressants at the time of diagnosis with depression, during, or following group IPT implementation.

### Acceptability

The refusal rate was 6.6% (2.6% refused to sign the consent form and 3.9% accepted to sign the consent form but did not attend any therapy sessions). The drop-out rate was 12.7%. Dropout occurred after attending the pre-group individual session (one patient) or at least three group sessions (8). The dropout was often justified by a specific reason (e.g., nonpsychiatric hospitalization [1]; leaving the country [1]; flooded home [1]; child custody issue [1]).

The quality of rendered service was rated as "excellent" or "good" by 92% of participants (Figure 1). The majority reported they received the kind of treatment they had expected (93%) and that group IPT met "all" or "most" of their needs (30 and 60%, respectively). Patient satisfaction with group IPT was high in 91% of participants, with 58% reporting they were "very satisfied" and 33% saying they were "satisfied."

### Feasibility

Main feasibility indicators are presented in Table 2. The majority of patients who completed group IPT (97%) had attended at least seven out of eight sessions. Three patients missed two sessions

**Table 1.** Characteristics of the sample

| Variables | N (%) |
|---|---|
| $N^{a}$ | 69 |
| Age (years) (median IQR) | 45 (39–53) (Min: 20–Max: 70) |
| Female | 35 (50.7) |
| Living alone | 30 (43.5) |
| Unemployed | 29 (42.0) |
| No sharing status (mis.: 4) | 18 (26.1) |
| Recent move | 17 (24.6) |
| Financial difficulties | 57 (82.6) |
| HIV infection duration (years) (median IQR) (mis.: 3) | 8 (2–13) (Min: 0,5–Max: 19) |
| Antiretroviral treatment adherence at baseline | 66 (95.6) |

*Note:* 71 PLWH agreed to participate in group IPT but 2 patients were subsequently excluded from data analysis due to incorrect completion in their consent to participate.
Abbreviations: IQR, interval interquartile; mis., missing data.
[a] Including patients who dropped out group IPT.

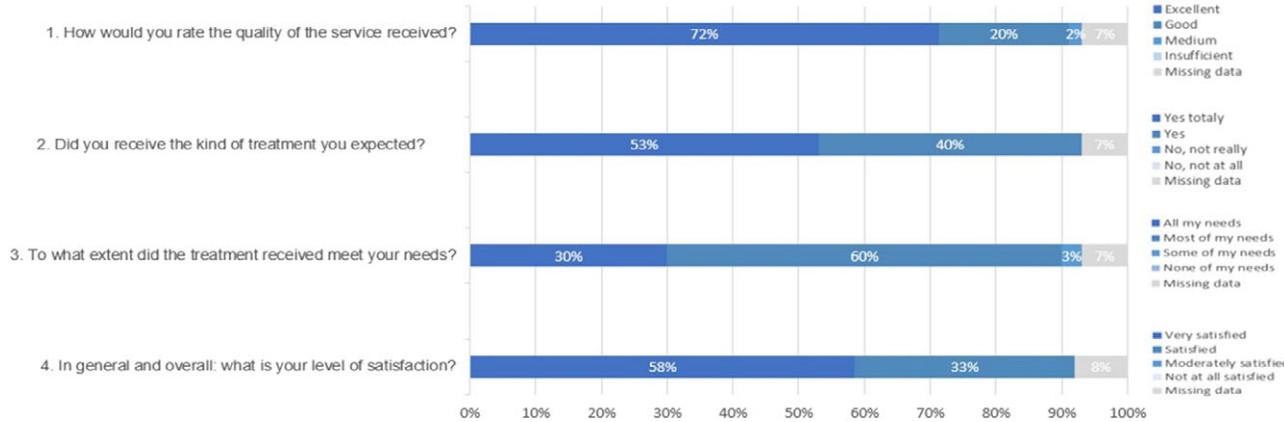

**Figure 1.** Patient satisfaction with their participation in group IPT (*N* = 60).

(3.3%), and 13 missed one session (21.7%). Missed sessions were justified (i.e., death in the family, illness, work constraints). At baseline, a number of PLWH who subsequently completed group IPT had reported suicidal ideation; specifically, 4 patients (6.7%) reported suicidal thoughts "more than half of the time" or "almost every day" whereas 19 patients (31.7%) reported suicidal thoughts "several days." In contrast, upon completion of group IPT, none reported suicidal thoughts "more than half of the time" or "almost every day"; and only 2 patients (3.6%) reported suicidal thoughts "several days." At their 3-month follow-up, only 5 patients (11.4%) reported suicidal thoughts "several days."

In the course of group IPT implementation, there were some delays due to COVID-19 lockdown and religious holidays (one group was interrupted by the COVID-19 lockdown, then resumed).

**Table 2.** Main feasibility indicators of group IPT in Senegal

|  | Median IQR or *n* (%) | Details |
|---|---|---|
| Attendance (≥7 out of 8 sessions) | 97 | One missed session: 13 (21.7%) Two missed sessions: 3 (3.3%) |
| Justified nonattendance | 100 | Missing sessions due to death in the family, illness, work constraints |
| Death per suicide | 0 |  |
| Hospitalization for imminent risk for suicide | 0 |  |
| Suicidal ideations | – | Positive evolution for patients experiencing suicidal thoughts |
| Antidepressant prescriptions | 0 |  |
| Duration of ITP treatment (weeks) | 7 (7–9) | Delays due to COVID-19 lockdown and religious holidays (one group was interrupted by the COVID-19 lockdown, then resumed) |
| Wait time before treatment start (weeks) | 5 (2–12) |  |
| Therapy recommendation to other patients | 100 | Recommendations made by patients and facilitators |

Abbreviations: IPT, interpersonal therapy; IQR, interquartile rate.

All group facilitators were "very satisfied" with the implementation of group IPT (Supplementary Figure S2) and were either "very satisfied" or "rather satisfied" with the length and frequency of sessions. They all reported that implementing group IPT in their health care routine was not complicated (67% "no, not really" and 33% "no, not at all"). A major difficulty, however, was compliance with group IPT session times; the workload remained nonetheless acceptable. The facilitators' feedback was mixed on confidentiality management (particularly concerning HIV status) and on patient weekly attendance (i.e., the treatment burden of coming weekly for 8 weeks and associated transportation costs). All the facilitators (100% "yes totally") and patients (85% "yes totally," 15% "yes") said they would recommend group IPT to other patients.

### Evolution of PHQ-9, WHODAS and HIV-related scores

At baseline, 22% of patients had mild depressive symptoms; 50% had moderate symptoms and 28% had moderately severe to severe symptoms (Figure 2). By the end of treatment, only 22% had depressive symptoms (20% mild and 2% moderate, respectively). Three months' post-treatment, 20% had depressive symptoms (17% mild and 3% moderate, respectively).

At baseline, the mean PHQ-9 score was 13 (SD = 4) (Table 3). At the end of group IPT, the mean score was 2 (SD = 3). Three months following group IPT, the mean score was 3 (SD = 3). Symptomatic improvement was statistically significant between baseline and completion of group IPT ($p < 0.001$) (Table 3). The evolution of scores was stable between group IPT completion and subsequent 3-month follow-up. However, 4 patients (7%) with no or mild depressive symptoms at the end of group IPT had moderate symptoms 3 months later, and 1 patient with moderate depressive symptoms at the end of group IPT did not show improvement 3 months later.

A significant improvement in WHODAS scores was observed between baseline and the end of group IPT ($p < 0.001$). However, there was no change in scores between the end of group IPT and 3-month follow-up. Concerning HIV-related stigma, a significant improvement was observed only in negative self-image scores in the time period extending from baseline to group IPT completion ($p < 0.001$).

### Discussion

To our knowledge, the present study constituted the first group IPT implementation in West Africa. The implementation appeared

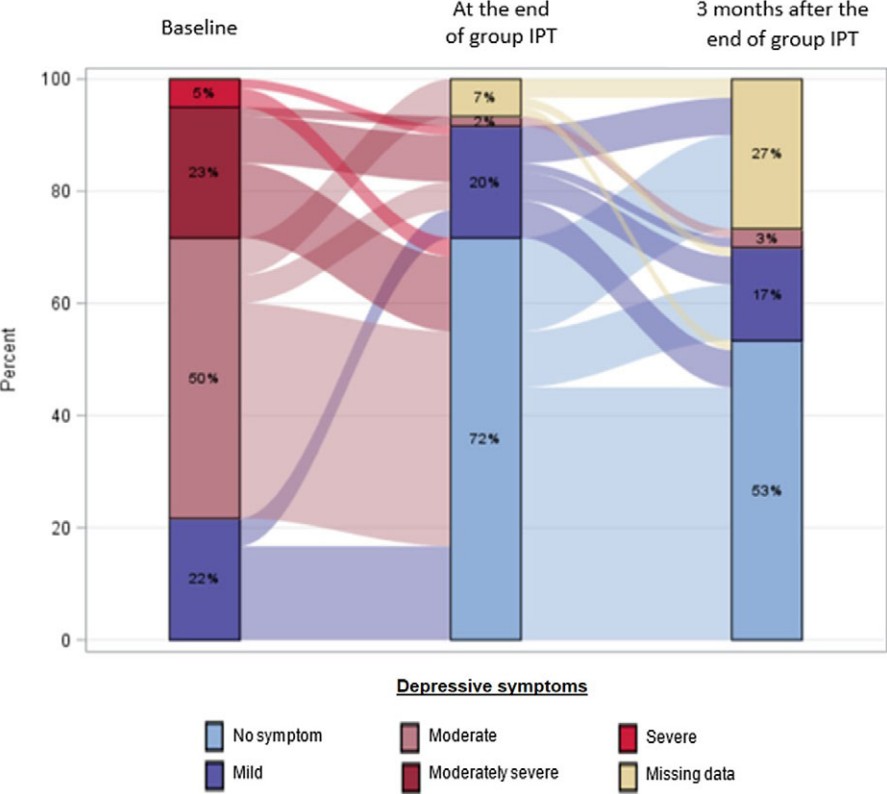

**Figure 2.** Description of the prevalence and the evolution of depressive symptoms according to severity, at baseline, end of group IPT and 3 months after the end of group IPT.

successful and promising in Senegal. Acceptability was high with low refusal rate, moderate drop-out rate and high patient satisfaction. Feasibility appeared good with high session attendance, satisfactory treatment duration and workload for group facilitators. All patients and all facilitators endorsed group IPT, with willingness to recommend it. On the other hand, identification of patients with depression (i.e., delay in group formation) proved challenging and treatment burden of coming weekly for 8 weeks could have been a barrier for some patients. In terms of outcome, patients who completed group IPT had a drastic decrease in depression severity and disability, but no significant reduction of HIV-related stigma except with regard to negative self-image.

Compared to other studies using group IPT, the present study produced comparable or better results in terms of session attendance and patient retention (Bolton et al., 2003; Petersen et al., 2012, 2014; Meffert et al., 2021). Data on the implementation of other interventions for treating depression among PLWH in sub-Saharan Africa are scarce. One study in Uganda reported promising results from a 8-week group support psychotherapy to treat depression in PLWH. Attrition was also minimal and the majority of the participant attended all the sessions (Nakimuli-Mpungu et al., 2014, 2020). However, direct comparisons remain difficult due to different study design. Good acceptability and feasibility may be attributed to various aspects of the study, including cultural adaptation

**Table 3.** Description of the PHQ-9, WHODAS and HIV-related stigma scores at each visit and comparisons of scores across visits

| N = 60 | Description of the scores at each visit | | | | | | Comparison of the score between | |
|---|---|---|---|---|---|---|---|---|
| | Baseline | | End of group IPT | | 3 months after group IPT | | Baseline and end of group IPT | End of group IPT and 3 months after group IPT |
| | Mean (SD) | Mis. | Mean (SD) | Mis. | Mean (SD) | Mis. | *p* | *p* |
| PHQ-9 scores | 13 (4) | 0 | 2 (3) | 4 | 3 (3) | 16 | **<0.0001** | 0.49 |
| WHODAS scores | 26 (6) | 0 | 16 (5) | 4 | 17 (6) | 16 | **<0.0001** | 0.78 |
| HIV-related Stigma subscores | | | | | | | | |
| Personalized stigma | 7 (2) | 8 | 7 (2) | 16 | 7 (2) | 24 | 0.76 | 0.92 |
| Disclosure concerns | 12 (1) | 0 | 12 (1) | 6 | 11 (1) | 17 | 0.39 | 0.79 |
| Concerns about public attitudes | 10 (2) | 0 | 10 (2) | 4 | 10 (1) | 16 | 0.98 | 0.45 |
| Negative self-image | 9 (2) | 0 | 7 (1) | 4 | 8 (2) | 16 | **<0.0001** | 0.39 |

Abbreviation: mis., missing data. *p*-values in bold indicate significant results (*p*<0.05).

(detailed elsewhere: Ziadeh et al., in press). First, the group modality seemed to resonate well with the collectivistic nature of many African cultures (Verdeli et al., 2003b; Nakimuli-Mpungu et al., 2014; Petersen et al., 2014), including Senegal's. In the context of HIV/AIDS and associated stigma, this is especially pertinent as groups can be powerful means to break social isolation and garner peer support. For PLWH, a group may be more effective than individual counseling (UNAIDS/WHO, 2004) and more helpful than friends and family support (Balmer, 1994). Second, the intervention was implemented within a task-shifting paradigm centered on clinical training and skill building (e.g., drawing on a clear and detailed manual, ongoing supervision), which was critical given that group facilitators had no prior mental health clinical experience. Task-shifting had proved successful in other studies (Dua et al., 2011; Patel et al., 2011; Du Zeying et al., 2022), and could increase treatment accessibility across a country, particularly in rural areas (Nakimuli-Mpungu et al., 2021). By the same token, it requires more trained staff and a self-sustaining training model (e.g., where facilitators-in-training become supervisors and trainers for future providers).

The present study showed that group IPT was associated with significant and sustainable reduction in depressive symptoms as well as improvement in functioning 3 months post-treatment, in line with other studies but conducted over a longer time period (Bolton et al., 2003; Meffert et al., 2021). Even if those findings need to be viewed with caution given the lack of a control group, they highlight that group IPT had positive impact on the lives of PLWH suffering with depression.

However, PLWH did not experience a positive impact on HIV-related stigma after group IPT (except for self-image). Active ingredients, such as coping skills, social support and behavioral activation, are important for effective mental health (Nakimuli-Mpungu et al., 2021) and are included in group IPT. In the context of high HIV-related stigma and as stigma impacts depressive symptoms (Nakimuli-Mpungu et al., 2014), resilience against stigma should also be considered an active ingredient and included as a component of mental health interventions.

We identified two main barriers to sustainability of group IPT in routine care: (1) difficulties in identifying patients with depression, which led to longer wait times for patients to join their group and, therefore, access treatment; and (2) treatment burden, especially for those living far away from Dakar, particularly due to transportation costs. The COVID-19 pandemic presented additional challenges. The study (conducted during the first 2 years of the pandemic) had to be paused for 4 months following a confinement order issued by the Senegalese government. This had a negative impact on the process of case identification. A number of PLWH balked at the idea of coming to their consultation appointments for fear of being infected with COVID-19 (The Global Fund, 2021). The frail ones might have stayed home, especially when depressed.

One implication for case identification – beyond the effects of a pandemic – is the necessity to systematize and optimize the process, especially that depression screening is not performed routinely in Senegalese HIV care centers. Previous studies conducted in Africa may offer some guidance. In Uganda, the community was involved in identifying patients and spreading information about group support therapy (i.e., they did not use group IPT) (Nakimuli-Mpungu et al., 2015). In Kenya, patients were recruited through referral or self-referral following informational talks in clinics' check-in and waiting areas (Meffert et al., 2021). These strategies may reduce stigma and increase voluntary screening.

Treatment burden is the requirement to attend group sessions over 8 weeks, once a week, which facilitators deemed a bit excessive in terms of time and money commitment, particularly for patients who were employed or who lived far away from the health center. Transportation cost has already been identified as a factor affecting access to treatment in LMICs (Petersen et al., 2014; Meffert et al., 2021). The dissemination of group IPT to different HIV care centers may help bring treatment closer to patients and reduce transportation costs.

In the context of implementation barriers, there is also the challenge of integrating group IPT into an already operational system, with preexisting tasks and procedures. In the present study, group IPT providers appeared to manage well in the context of task-shifting, as evidenced by their reported satisfaction with the workload and the intervention. Nevertheless, long-term adverse consequences (e.g., worker dissatisfaction, burnout, attrition) cannot be ruled out without a dedicated study.

The present study showed a significant reduction in depressive symptoms and some stability in depression scores post-treatment, suggesting that an 8-week intervention might be sufficient to produce long-term results. On the other hand, for patients who no longer improved or whose condition worsened at the 3-month follow-up, the 8-week intervention might have been too short. We hypothesized that a brief protocol would not permit all patients to develop the necessary skills to get to remission and, in the case of some patients, treatment should be extended. Previous studies with good treatment outcomes (e.g., Uganda study) have typically used a 16-week treatment protocol (Bolton et al., 2003). And, in treatment planning, longer interventions may be favored. In this vein, counselors in an Ethiopian study wished to increase the number of sessions (Asrat et al., 2021). On the other hand, increasing the number of sessions can pose a different set of challenges, such as more workload and higher demands on resources, and thus undermine treatment acceptability. In effect, lower treatment attendance can be observed in studies using longer treatment protocols. For example, a 12-week intervention implemented in South Africa (Petersen et al., 2012) had initially high attendance, but only 50% of patients completed 11–12 treatment sessions.

Apparently, group IPT continues to be helpful beyond the termination session, in that group members go on helping and supporting one another. In Uganda, informal group meetings were reported beyond the 16 weeks of group IPT (Bass et al., 2006). Seemingly, patients used this time for informal counseling, discussing how to tackle their problems and they felt socially and emotionally (Bass et al., 2006). For this reason, perhaps, treatment duration may be less important than cultivating a group culture that is supportive and self-sustaining.

As structural realities in LMICs encourage the use of adapted, brief, structured psychological interventions (Patel et al., 2007), one needs to keep in mind the individual needs of patients. And though group IPT has proved generally helpful to PLWH suffering with depression, the therapeutic scheme and modality still need to be adapted to the patient's specific needs. For example, individual IPT could be used following group IPT, if a patient expresses the need for it.

### Limitations

The present study had a number of limitations. First, the specificity and particularities of the setting – that is, a hospital at the top of the health pyramid, with a dedicated psychiatry department – could have limited the generalizability of these findings. For this reason,

the results need to be confirmed in other contexts. Second, some aspects of the acceptability and feasibility were assessed with quantitative measures. In this context, social-desirability bias cannot be ruled out. Third, the feasibility of the intervention from the facilitators' point of view was only described but not statistically assessed given the limited number of participating staff. Fourth, some aspects of the feasibility (i.e., patients' experience with group IPT; facilitators' perception of task-shifting; treatment setting; possible difficulties encountered during the sessions) could not be evaluated in the present study but will be assessed with qualitative interviews in a future study.

## Conclusion

Group IPT, conducted in line with the WHO manual and within a task-shifting paradigm, to treat depression in PLWH, showed good acceptability and feasibility and emerged as a promising intervention that could close the mental health treatment gap in Senegal. The intervention's group modality appeared suitable for Senegal's culture (i.e., breaking patient isolation and enabling peer-to-peer exchanges). Identified barriers to treatment may be overcome with specific measures, including (1) bringing services closer to patients and making group IPT available beyond Dakar; (2) improving current patient identification procedures and (3) adapting the therapeutic scheme to the specific needs of patients. Such measures may transform group IPT into a standard of care for PLWH with depression across Senegal and beyond.

**Open peer review.** To view the open peer review materials for this article, please visit http://doi.org/10.1017/gmh.2023.31.

**Supplementary material.** The supplementary material for this article can be found at https://doi.org/10.1017/gmh.2023.31.

**Data availability statement.** The data that support the findings of our study are available from the corresponding author upon reasonable request.

**Acknowledgments.** The authors would like to thank the IeDEA West Africa region: Site investigators and cohorts: *Adult cohorts*: Marcel Djimon Zannou, CNHU, Cotonou, Benin; Armel Poda, CHU Souro Sanou, Bobo Dioulasso, Burkina Faso; Fred Stephen Sarfo, Komfo Anokeye Teaching Hospital, Kumasi, Ghana; Eugene Messou, ACONDA CePReF, Abidjan, Cote d'Ivoire; Henri Chenal, CIRBA, Abidjan, Cote d'Ivoire; Kla Albert Minga, CNTS, Abidjan, Cote d'Ivoire; Emmanuel Bissagnene and Aristophane Tanon, CHU Treichville, Cote d'Ivoire; Moussa Seydi, CHU de Fann, Dakar, Senegal; Akessiwe Akouda Patassi, CHU Sylvanus Olympio, Lomé, Togo. *Pediatric cohorts*: Sikiratou Adouni Koumakpai-Adeothy, CNHU, Cotonou, Benin; Lorna Awo Renner, Korle Bu Hospital, Accra, Ghana; Sylvie Marie N'Gbeche, ACONDA CePReF, Abidjan, Ivory Coast; Clarisse Amani Bosse, ACONDA_MTCT+, Abidjan, Ivory Coast; Kouadio Kouakou, CIRBA, Abidjan, Cote d'Ivoire; Madeleine Amorissani Folquet, CHU de Cocody, Abidjan, Cote d'Ivoire; François Tanoh Eboua, CHU de Yopougon, Abidjan, Cote d'Ivoire; Fatoumata Dicko Traore, Hopital Gabriel Toure, Bamako, Mali; Elom Takassi, CHU Sylvanus Olympio, Lomé, Togo; *Coordinators and data centers*: François Dabis, Renaud Becquet, Charlotte Bernard, Shino Chassagne Arikawa, Antoine Jaquet, Karen Malateste, Elodie Rabourdin and Thierry Tiendrebeogo, ADERA, Isped & INSERM U1219, Bordeaux, France. Sophie Desmonde, Julie Jesson and Valeriane Leroy, Inserm 1027, Toulouse, France. Didier Koumavi Ekouevi, Jean-Claude Azani, Patrick Coffie, Abdoulaye Cissé, Guy Gnepa, Apollinaire Horo, Christian Kouadio and Boris Tchounga, PACCI, CHU Treichville, Abidjan, Côte d'Ivoire.

**Author contribution.** C.B., N.R., F.D., M.S. and N.F.N. designed the study and wrote the protocol. C.B. and H.F. managed and analyzed the data. C.B. wrote the first draft of the manuscript. J.M.T. and A.D. realized the inclusion of the patients and collected the data under the supervision of M.S. and N.F.N. I.N. and O.S. made the diagnosis of depression and supervised the team, as referring psychiatrist. S.Z., H.V., T.B. and L.J. as experts in IPT, trained the team and S.Z. supervised group IPT throughout the project. They helped in the analyses of the data. H.F., N.R. and S.Z. critically reviewed the early draft of the manuscript. All authors critically reviewed and have approved the final manuscript.

**Financial support.** This work was supported by the National Institute of Mental Health (NIMH), National Cancer Institute (NCI), the Eunice Kennedy Shriver National Institute of Child Health & Human Development (NICHD) and the National Institute of Allergy and Infectious Diseases (NIAID) of the U.S. National Institutes of Health (NIH), as part of the International Epidemiologic Databases to Evaluate AIDS (IeDEA) under Award Number U01AI069919. The content is solely the responsibility of the authors and does not necessarily represent the official views of the National Institutes of Health.

**Competing interest.** The authors declare that they have no conflicts of interest.

**Ethics statement.** This work complies with the ethical standards of the relevant national and institutional committees on human experimentation and with the Helsinki Declaration of 1975, as revised in 2008.

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
