## [Reviewer Report]

Bordeaux

01/06/2023

Dear Editor,

Please find the attached manuscript entitled “Management of depression in people living with HIV in Senegal: acceptability and feasibility of group interpersonal therapy” that we submit to Cambridge Prisms: Global Mental Health for consideration as a Research Article. 

Depression is highly prevalent in people living with HIV (PLHIV), and has negative consequences for daily life and care. But in Sub-saharan Africa, depression remains often underdiagnosed and undertreated. In this context, in resource-limited settings, the World Health Organization (WHO) recommends task-shifting and psychotherapeutic treatments to treat depression. We evaluated for the first time the acceptability and feasibility of group interpersonal therapy (group IPT), combined with a task-shifting approach, to treat depression in PLHIV in Senegal. Sixty patients completed group IPT. Groups were facilitated by trained social workers and lay workers and following the WHO manual. With pre and post-treatment evaluations, including a 3-month follow-up, we also evaluated the evolution of depressive symptoms severity, related disability and stigma. Our results showed that group IPT showed good acceptability and feasibility to treat depression in PLHIV in Senegal. group IPT also drastically decreased depression severity and associated disability. We also identified barriers to treatment that are essential to ensure sustainability of this therapy in Senegal. We highlighted that group IPT emerged as a promising intervention that could help to reduce the mental health treatment gap in Senegal. We also discussed how some measures might transform group IPT into a standard of care for depressed PLHIV across Senegal and beyond.

As a Research Article, the content of this submission includes one abstract, one full-text document, three figures, two tables, and a supplementary data file. We certify that all co-authors have seen and approved this manuscript’s content and that it is not under review at any other journal. The authors have no financial interest to declare.

We hope this manuscript will be of interest to your readers and respond to the criteria for publication in Cambridge Prisms: Global Mental Health.

Sincerely yours,

Bernard Charlotte, PhD

Team Global Health in the Global South,

University of Bordeaux, INSERM UMR 1219, IRD EMR 271, BPH, 

146 rue Léo Saignat, 33076 BORDEAUX Cedex (FRANCE) 

Tél: 05 57 57 56 65, Fax: 05 56 24 00 81

Mail: charlotte.bernard@u-bordeaux.fr

---

## [Reviewer Report]

Thank you for the opportunity you offered me to read and review the article entitled “Management of depression in people living with HIV in Senegal: acceptability and feasibility of group Interpersonal Therapy”. 

Overall comment

The article presented the required information expected from a research article coherently. The research area and its focus on a special population are much needed in the literature. Although the article has several strengths, it needs further revision and revisions to be publishable. 

Abstract: measurement is unclear, primary outcomes are unknown and the method of analysis needs to be explained clearly. 

Introduction: it is well presented with the narration of the current studies. However, it lacks scientific argument and justification on the therapeutic mechanisms of interpersonal therapy and why interpersonal therapy could be beneficial for people with HIV/AIDS.

Methods: the study design is blurred, the study population need to be described adequately, and the training and competency level of intervention providers are unclear. Outcome measurement needs more information such as the cultural validation of tools is unknown, particularly in the HIV population. The assessment of acceptability and feasibility looks mixed up, with inadequate clarity. Both would have been good to be assessed qualitatively as well. The descriptive statistics using percentiles would be challenging for readers to compare the change before and after the intervention. it was excellent if variables were analyzed as a continuous outcome rather than a binary outcome. More importantly, no information regarding data collection methods, tools, and timing/points of the data collection period. 

Results: it is more of a detail of information that needs more synthesis. The result would be clear if it was assessed through qualitative evaluation and effect size estimations. 

Discussion: it is clearly presented but still needs strong narration, argument and justifications.

Specific questions

Abstract – what is the evidence to say “high satisfaction”?

Methods – what guideline/manual the psychiatrists used to confirm depression? how French psychiatrists could have the cultural competency to diagnose depression? Why the local psychiatrists were not chosen? What do you mean “gender-specific?

Results: why you excluded one patient because he/she has taken individual IPT and two others because they are residing far from the intervention setting? Because these were not mentioned in the exclusion criteria.

---

## [Reviewer Report]

REVIEWWER’S COMMENTS. MANUSCRIPT: GMH-23-0004

TITLE: Management Of Depression In People Living With HIVIn Senegal: Acceptability And Feasibility Of Group Interpersonal Therapy.

GENERAL COMMENT

This is a good paper, relevant, topical and well written. The authors are well versed with the relevant literature. The methodology was well described although the study design was not succinctly articulated. The method was described but lacked a clear description of the study setting and how the study participants were recruited. Data protection was not mentioned. Statistical analysis was well described. The results were fairly well described but the tables needed to be presented more clearly. All in all, it is a good paper that needs just a few minor adjustments.

SPECIFIC COMMENTS

Tilte: This should state “….people living with HIV/AIDS….”

Abstract: There are two abstracts with two sets of Key words: 

The first set says (page 1 in the box): Depression, HIV, Group support psychotherapy, Developing countries.

The second set says (Line 19): Depression, HIV, Sub-Saharan Africa, Group Interpersonal Therapy.

The first set is incorrect. This paper is about Group Interpersonal Therapy in Sub Saharan Africa, more specifically Senegal, and not in any other developing countries in other continents. Also it confuses “Group Interpersonal Therapy” with “Group support psychotherapy”. These are two distinctly different therapies developed by different researchers at different times in different continents. The second abstract is the correct one in all items.

Background: This was well written and to the point. The objects were clearly stated.

Methods: Study design (Lines 81-87): The study design is not clearly articulated as a cross-sectional descriptive study comparing pre- and post-intervention data on a number of specified parameters when Group Interpersonal Therapy was used to treat depression in PLHIV (or PLWH). The authors need to state that the study instruments were translated in Wolof and by whom and how this translation was done. 

Lines 90-95: The study site was described as Fann National University Hospital Center, but what was the actual study setting? Was it in the Department of Psychiatry? Or in the two services units which provided treatment and follow-up for PLHIV i.e. the Infectious and Tropical Diseases unit and the Outpatient Treatment Center? Were all the study participants outpatients or were there some inpatients? How were they approached and recruited for the study? The study population was PLHIV but what was the accessible population and how were the study participants actually recruited (the study unit) and consented? Privacy? Confidentiality? Data protection.

Line 131: “….over the span of several months”. Please state the exact number of months.

Results: The presentation of the results needed more clarity. 

Line 227: Indicate that Table 1 should be inserted here as Table 1: Characteristics Of The Sample.

Line 237: Acceptability: Indicate that Table 2 goes here as Table 2: PHQ-9, WHODAS and HIV-related stigma scores at Baseline, End of IPT and 3-months after IPT.

Lines 249-273: Feasibility. This was described verbatim with no table. However “Feasibility” is the gist of the study as indicated in the Title. For better clarity, these results needed to be indicated in tabular format followed by the verbatim description. 

Discussion: The authors need to avoid falling into the trap of describing changes in symptomatology as indicating feasibility e.g. improved depression scores. Symptom improvement is a measure of effectiveness or efficacy of an intervention and not its feasibility. Feasibility is measured by how practically possible one can implement an intervention; e.g. number of sessions attended by the participants; numbers of participants who actually attended and their retention in therapy etc. Lastly in the discussion, the authors seemed to confuse Group Interpersonal Psychotherapy (Group IPT) with Group Support Psychotherapy (GSP) in their citations as if they were one and the same. Group IPT and GSP are two distinctly different therapies developed by different researchers at different times in different continents. However they are both efficacious.

The conclusions, recommendations and references are adequate.

---

## [Reviewer Report]

Dear Authors - thank you for this manuscript which two other reviewers and I have read. We conclude the paper is well-written on a topic of great importance and could benefit from increased precision and explanation throughout. Therefore, I invite you to revise and resubmit this manuscript to address the reviewers' queries.

In addition to the reviewers, I also have a few comments:

1. Why was IPT chosen and not another low-intensity depression intervention (for example, PM+, also recommended by the WHO)? What conditions, factors, considerations, etc. were taken into account when choosing IPT?

2. The manuscript could be strengthened (Intro, Discussion) by comparing your findings with other depression interventions for PLWH. What is the MH treatment gap among PLWH?

3. I recommend consistent use of “PLWH with depression” rather than “depressed PLHIV” (e.g., lines 7, 228). The former is “people first language,” whilst the latter places the disease state first.

4. In the results, avoid using terms that are suggestive of opinion/interpretation, e.g., avoiding words like “only,” “never showed up,” and “non-conformity” -- the latter, especially, could be interpreted as judgmental. In line 202, for example, “but only 165” could be replaced with “and.”

5. In line 301 “patient transportation costs” -- this appears to be novel data, not explicitly addressed in the results. As such, either provide the results that support this statement or remove this statement.

6. Line 344 -- I believe the word meant is “balked” (rather than “bulked”)

---

## [Reviewer Report]

Bordeaux

01/06/2023

Dear Editor,

Please find the attached manuscript entitled “Management of depression in people living with HIV in Senegal: acceptability and feasibility of group interpersonal therapy” that we submit to Cambridge Prisms: Global Mental Health for consideration as a Research Article. 

Depression is highly prevalent in people living with HIV (PLHIV), and has negative consequences for daily life and care. But in Sub-saharan Africa, depression remains often underdiagnosed and undertreated. In this context, in resource-limited settings, the World Health Organization (WHO) recommends task-shifting and psychotherapeutic treatments to treat depression. We evaluated for the first time the acceptability and feasibility of group interpersonal therapy (group IPT), combined with a task-shifting approach, to treat depression in PLHIV in Senegal. Sixty patients completed group IPT. Groups were facilitated by trained social workers and lay workers and following the WHO manual. With pre and post-treatment evaluations, including a 3-month follow-up, we also evaluated the evolution of depressive symptoms severity, related disability and stigma. Our results showed that group IPT showed good acceptability and feasibility to treat depression in PLHIV in Senegal. group IPT also drastically decreased depression severity and associated disability. We also identified barriers to treatment that are essential to ensure sustainability of this therapy in Senegal. We highlighted that group IPT emerged as a promising intervention that could help to reduce the mental health treatment gap in Senegal. We also discussed how some measures might transform group IPT into a standard of care for depressed PLHIV across Senegal and beyond.

As a Research Article, the content of this submission includes one abstract, one full-text document, three figures, two tables, and a supplementary data file. We certify that all co-authors have seen and approved this manuscript’s content and that it is not under review at any other journal. The authors have no financial interest to declare.

We hope this manuscript will be of interest to your readers and respond to the criteria for publication in Cambridge Prisms: Global Mental Health.

Sincerely yours,

Bernard Charlotte, PhD

Team Global Health in the Global South,

University of Bordeaux, INSERM UMR 1219, IRD EMR 271, BPH, 

146 rue Léo Saignat, 33076 BORDEAUX Cedex (FRANCE) 

Tél: 05 57 57 56 65, Fax: 05 56 24 00 81

Mail: charlotte.bernard@u-bordeaux.fr

---

## [Reviewer Report]

This revised version is much better. The authors addressed the reviewer’s comments. On Line 469, there seems to be a missing word....“they felt socially and emotionally supported...” On Lines 452-454, for the participants who did not improve or worsened by three months, would it not be ethically prudent to suggest/recommend adding an antidepressant to their treatment instead of just a longer period of IPT ? Indeed Patel et al (2007) on Line 427 recommended some individual specifics for some patients.